# Young people's data governance preferences for their mental health data: MindKind Study findings from India, South Africa, and the United Kingdom

Solveig K. Sieberts[1‡], Carly Marten[1‡], Emily Bampton[2], Elin A. Björling[3], Anne-Marie Burn[4], Emma Grace Carey[4], Sonia Carlson[1], Blossom Fernandes[2], Jasmine Kalha[5], Simthembile Lindani[6], Hedwick Masomera[6,7], Lakshmi Neelakantan[2], Lisa Pasquale[1], Swetha Ranganathan[5], Erin Joy Scanlan[1], Himani Shah[5], Refiloe Sibisi[8,9], Sushmita Sumant[5], Christine Suver[1], Yanga Thungana[6], Meghasyam Tummalacherla[1], Jennifer Velloza[10,11], Pamela Y. Collins[11,12], Mina Fazel[2], Tamsin Ford[4,13], Melvyn Freeman[14,15], Soumitra Pathare[5], Zukiswa Zingela[7], The MindKind Consortium[¶], Megan Doerr[1]*

1 Sage Bionetworks, Seattle, Washington, United States of America, 2 Department of Psychiatry, University of Oxford, Oxford, Oxfordshire, United Kingdom, 3 Human Centered Design and Engineering, University of Washington, Seattle, Washington, United States of America, 4 Department of Psychiatry, University of Cambridge, Cambridge, Cambridgeshire, United Kingdom, 5 Centre for Mental Health Law & Policy, Indian Law Society, Pune, Maharashtra, India, 6 Department of Psychiatry, Walter Sisulu University, Eastern Cape, South Africa, 7 Nelson Mandela University, Gqeberha, Eastern Cape, South Africa, 8 Activate Change Drivers ZA, Johannesburg, Gauteng, South Africa, 9 University of Johannesburg, Johannesburg, Gauteng, South Africa, 10 Department of Epidemiology & Biostatistics, University of California San Francisco, San Francisco, California, United States of America, 11 Department of Global Health, University of Washington, Seattle, Washington, United States of America, 12 Department of Psychiatry and Behavioral Sciences, University of Washington, Seattle, Washington, United States of America, 13 Cambridgeshire and Peterborough Foundation NHS Trust, Fulbourn, Cambridgeshire, United Kingdom, 14 University of Stellenbosch, Stellenbosch, Western Cape, South Africa, 15 Higher Health, Centurion, Gauteng, South Africa

‡ SKS and CM are joint first authors on this work.
¶ Membership of the MindKind Consortium is provided in the Acknowledgments.
* megan.doerr@sagebionetworks.org

**Data Availability Statement:** Data from the quantitative arm are available through Synapse (www.synapse.org/MindKind). This includes the

## Abstract

Mobile devices offer a scalable opportunity to collect longitudinal data that facilitate advances in mental health treatment to address the burden of mental health conditions in young people. Sharing these data with the research community is critical to gaining maximal value from rich data of this nature. However, the highly personal nature of the data necessitates understanding the conditions under which young people are willing to share them. To answer this question, we developed the MindKind Study, a multinational, mixed methods study that solicits young people's preferences for how their data are governed and quantifies potential participants' willingness to join under different conditions. We employed a community-based participatory approach, involving young people as stakeholders and co-researchers. At sites in India, South Africa, and the UK, we enrolled 3575 participants ages 16–24 in the mobile app-mediated quantitative study and 143 participants in the public deliberation-based qualitative study. We found that while youth participants have strong preferences for data governance,

Participant preferences (https://doi.org/10.7303/syn51225257), Participant acceptability (https://doi.org/10.7303/syn51225253) and Participant votes (https://doi.org/10.7303/syn51225260) data. Code for the enrollment website is available through Github (https://github.com/Sage-Bionetworks/GlobalMentalHealthDatabank). Extended quotes from the qualitative arm are available in the Supporting Results. The materials used to inform participants prior to public deliberation sessions are available at https://doi.org/10.7303/syn35371551.

**Funding:** The MindKind Study was commissioned by the Mental Health Priority Area at Wellcome Trust (https://wellcome.org/) from Sage Bionetworks (LMM and MD). The funders had no role in study design, data collection and analysis, decision to publish, or preparation of the manuscript.

**Competing interests:** The authors have declared that no competing interests exist except for author Tamsin Ford. Tamsin Ford declares: I have read the journal's policy and have the following competing interests: I consult to Place2Be, a third sector organization providing mental health support to children, parents and staff in Schools, and am the Vice Chair of the Association of Child and Adolescent Mental Health.

these preferences did not translate into (un)willingness to join the smartphone-based study. Participants grappled with the risks and benefits of participation as well as their desire that the "right people" access their data. Throughout the study, we recognized young people's commitment to finding solutions and co-producing research architectures to allow for more open sharing of mental health data to accelerate and derive maximal benefit from research.

## Introduction

Unprecedented opportunities to better understand behavioral and emotional trajectories are now available to researchers from the sheer quantity of data that can be collected through mobile devices. Unlike in-person clinical data collection, remote research through smartphone apps on mobile devices can collect frequent, longitudinal data about lived experience directly from young people, given how readily this generation has embraced the use of smartphone devices [1, 2]. Three quarters of people affected by lifelong mental illness, including depression and anxiety, experience their first episode before the age of 24 [3], therefore the need to harness this data is even more important. However, data pertaining to mental health and lifestyle are considered highly personal, and efforts toward collection and analysis need to be responsive to the privacy concerns of participants. Very little is known about the data governance preferences of young people and their effect on willingness to participate in research. As the research community has increasingly embraced the importance of data sharing to accelerate and receive maximal value from research [4], data governance has tended to be crafted based on input from researchers but not participants [5]. As we continue to appreciate the shortcomings of often exclusionary, possibly exploitative, and paternalistic approaches to research, more inclusive, participant-driven approaches to data governance are needed [6–8].

This paper describes the *MindKind Study*, a program of work to understand the data governance preferences of young participants (aged 16–24 years) in smartphone-based mental health research. This work was co-produced with young people by grounding the study in participatory research methodology, involving youth stakeholders and researchers as equal partners [9]. The *MindKind* Study employed a mixed methods approach, pairing a quantitative, smartphone-based study arm with a qualitative, public deliberation study arm to investigate the participatory behaviors, concerns, and desires of young people with respect to digital mental health research. These inquiries were centered around a seven question governance typology based on previous conceptual research [10], which was further developed and customized for this study with input from youth and expert research panels.

This work was part of a larger effort to prototype and test the feasibility of building a large-scale global mental health databank for rich, longitudinal, electronically-derived mental health data from youth with a focus on the approaches, treatments, and interventions potentially relevant to anxiety or depression [11]. We conducted this study in three countries: India, South Africa, and the United Kingdom (UK), which were chosen for their range of economic, socio-cultural, and regulatory landscapes. In this paper we present participant's data governance preferences, acceptability, and thematic findings for sharing mental health data.

## Materials and methods

### Youth co-production

We endeavored to engage young people with lived experience of mental health challenges in all aspects of the MindKind Study. To this end, each site embedded a full time Professional

Youth Advisor (PYA) within their study team. The PYA worked closely with a site-specific Young People's Advisory Group (YPAG) which met bi-monthly throughout the project, enabling them to function as a consultative resource for the study team at every stage of the work, from study design and data collection to analysis and dissemination. Each YPAG was composed of 12–16 young people, and PYAs sought to ensure maximal heterogeneity within the groups with respect to geographic region, race/ethnicity, gender, languages spoken, and lived experience of mental health concerns. Study teams also worked with community partner organizations to ensure the representation of YPAG members from marginalized social groups. YPAGs were supplemented by an ad hoc International Youth Panel convened by study team members at the University of Washington. To separately assess the perspective of research and technology professionals, we assembled a Data Use Advisory Group (DUAG) to pilot and test project materials. See The MindKind Consortium, 2022 [11] for more details about the activities of the YPAGs and DUAG.

### Data governance typology

Based on prior work [10] codifying patterns of data governance in research literature, we developed a seven question data governance typology (Fig 1). This typology was created and iterated based on feedback from DUAG members and other big data technologists in our networks.

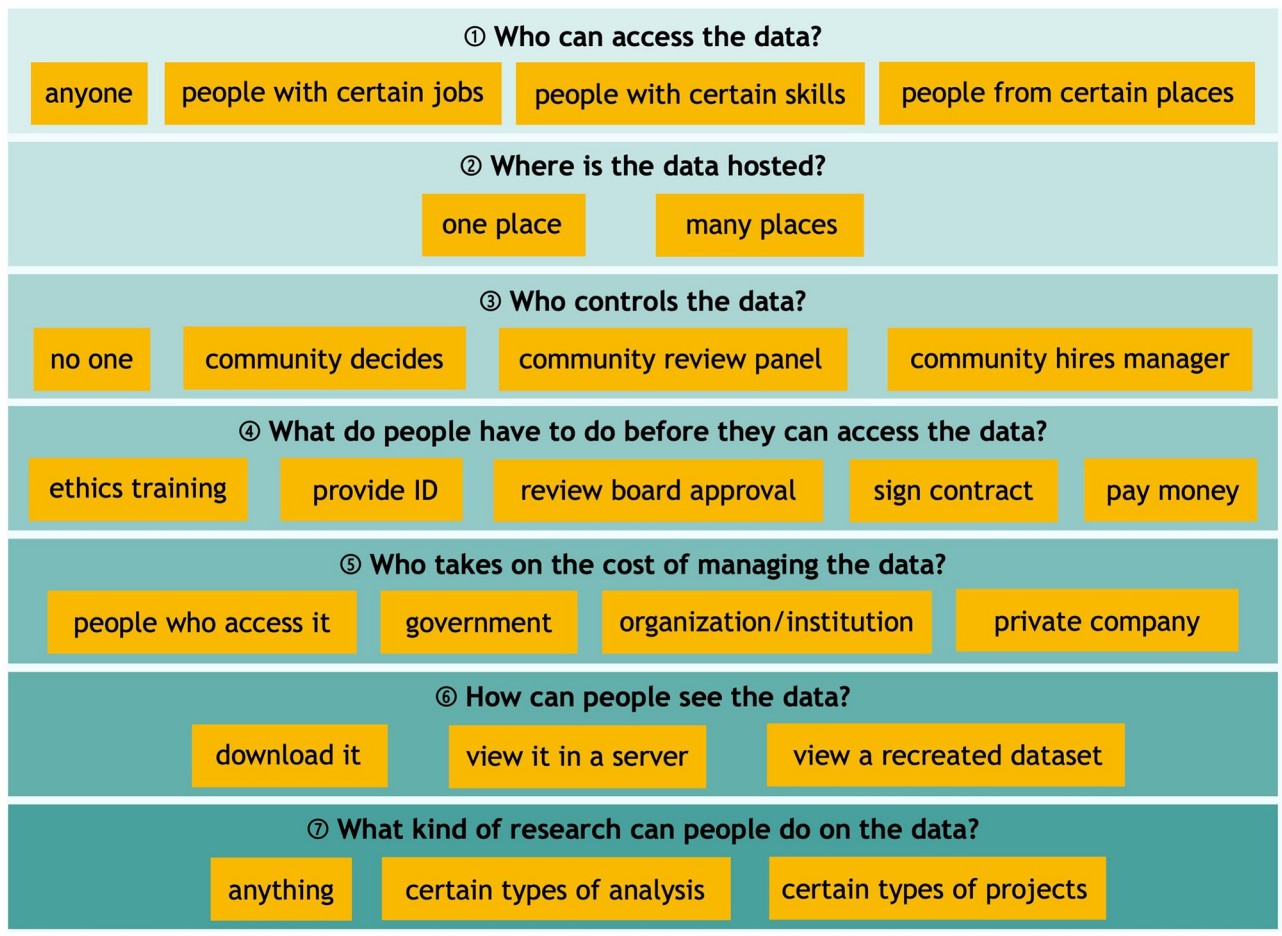

**Fig 1. Data governance typology.**

We then adapted it for language accessibility; for example, rather than stating that data could be hosted in a "centralized repository" and "federated storage," we used *one place* and *many places*, respectively. We asked YPAG and DUAG members for their preferences within this typology, investigating the tension between the privacy expectations of participants and data availability expectations of researchers. Areas of the greatest discordance between YPAG and DUAG members (questions 3, 6, and 7) formed the basis for test conditions in the quantitative study arm. All seven questions were asked of participants in the qualitative study arm.

## Ethics

The MindKind Study was approved by the relevant Institutional Review Boards and Ethics Boards in the US (WIRB #20212067), UK (University of Cambridge—Department of Psychology Research Ethics Committee: Ref. PRE.2021.031 and University of Oxford: Ref. R73366/RE00), South Africa (Walter Sisulu University #029/2021 and the Department of Higher Education and Training), India (India Law Society #ILS/242/2021), and by the Health Ministry Screening Committee (HMSC) in India.

## Recruitment

Quantitative study arm recruitment relied on direct outreach to youth (i.e., in person, peer recruitment, email, SMS), social media posts (i.e., on Twitter, Facebook, LinkedIn, Reddit), networking through local organizations (e.g., local news outlets), poster campaigns, radio advertisements, and targeted social media advertisements (e.g., Instagram, Facebook). Social media advertisements were responsible for the recruitment of the vast majority of quantitative study arm participants: 80%, 55%, and 90% of those recruited in India, South Africa, and the UK, respectively.

## Eligibility

To participate in the MindKind Study, youth participants lived in one of the participating countries and were legally able to provide consent (age 16–24 years in the UK or 18–24 years in India and South Africa). For quantitative substudy participation, youth were eligible if they could follow study instructions, read and understand English, and had access to an Android mobile phone. For qualitative substudy participation, youth were eligible if they could read and speak English in South Africa and the UK, could speak English or one of three regional languages in India, and had access to a device connected to the internet.

## Quantitative study arm

**Aims.**   The aims of the quantitative study arm were to understand (1) preferences of study participants regarding governance of their data and how it is accessed and (2) if governance policies affect study enrollment.

**Procedure.**   Potential participants were directed to an enrollment website (www.mindkindstudy.org), where they could access the website-based informed consent (eConsent) [12, 13]. Following eligibility checking, participants created a study account. They were then randomized, in equal proportions, to one of four different governance models–Group A, B, C, or D–designed to explore their preference for data governance models with varying levels of control over who can access the data and for what purpose, as well as to understand whether preferences influence enrollment (Fig 2). Group A assessed data governance preferences, while the remaining three (Groups B, C, D) addressed acceptability of various predefined data governance models.

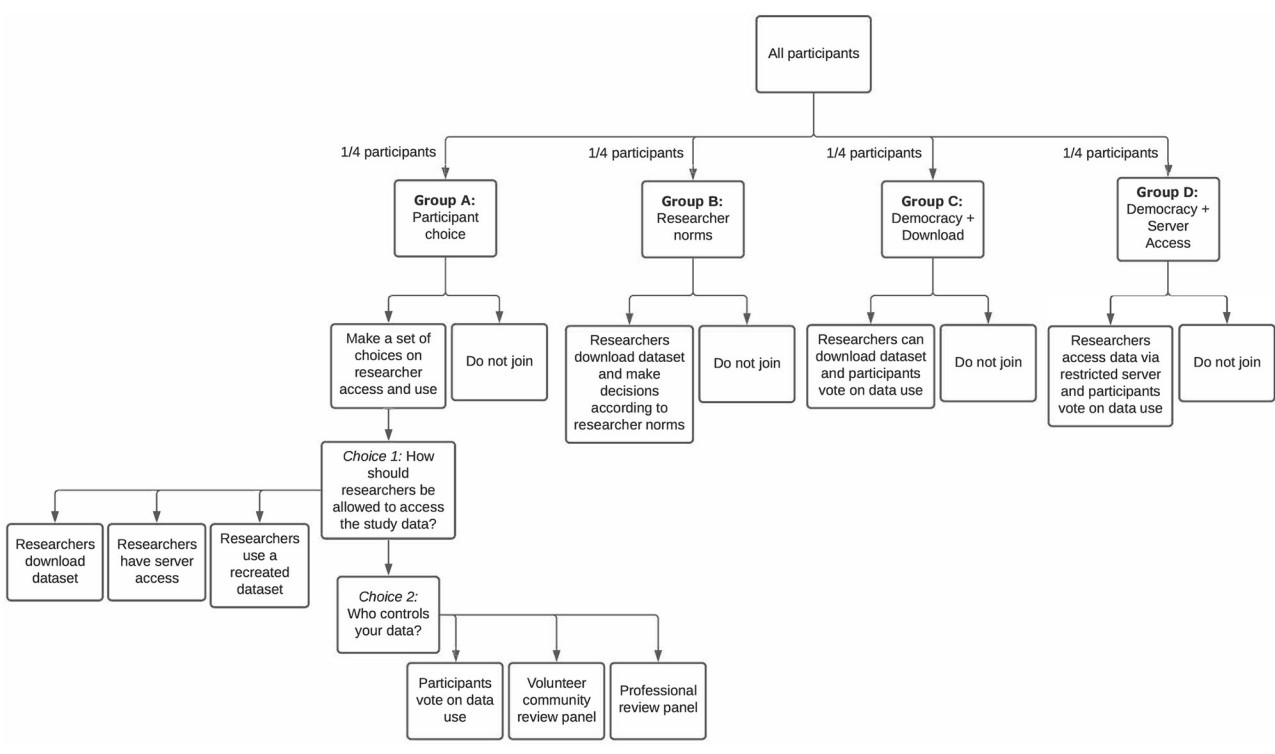

**Fig 2. Quantitative study design.**

Participants randomized to Group A were then prompted to select how researchers would be allowed to access their data and who would control access to the data. The options presented to participants were informed by the disparate preferences expressed by youth co-researchers versus the DUAG during study design. Group A participants responded to the following two questions.

Choice 1: How should researchers be allowed to access the study data?

1. Researchers should be allowed to download a copy.

2. Researchers should only be allowed to access the data in a secure server.

3. Researchers should only be allowed to see a recreated data set, not the real data. If researchers want to study the real data set, they have to ask the data steward to run their analysis for them. The steward only gives the researcher back the result, not the data.

Choice 2: Who controls the data?

1. Participants vote on data use (i.e., democracy): study participants who select this option get to vote on how the data is used, and the most popular terms are applied to all data regardless of how an individual votes. The results of the vote are shared with participants before data are shared. Any participant who disagrees with the vote may withdraw from the study.

2. Volunteer community review panel: participants selecting this option may choose to volunteer to serve as a data use request reviewer, taking one-week turns in this role on a rotating basis. Researchers will submit a statement telling the reviewers why they want to use the data. The reviewers will apply a set of criteria to decide yes or no. These criteria will be determined in advance by the whole group of volunteer reviewers.

3. Professional review panel: a paid panel will review data requests. This panel is a group of participants paid by the funder of the databank and may include research professionals (e.g., research ethics professionals). As above, researchers will have to submit a statement telling the reviewers why they want to use the data. The reviewers would decide yes or no, based on a set of criteria which will be developed in advance by the group.

Participants who were randomized to Group A were asked to select their data governance choices prior to providing consent.

In order to assess the acceptability of current governance standards relative to those that give participants a greater voice regarding how data are accessed and used, participants randomized to Groups B, C, or D were presented with a pre-specified governance model selected to test (1) whether democratic determination of data terms improves enrollment over current researcher-driven norms, and (2) whether limiting data access to a restricted server further improves enrollment. Specifically, the three models were:

- Research norm (Group B): This option presents current researcher community norms for data use, whereby researchers will be able to download a copy of the data from the databank following strict data security rules. Data may be used, unrestricted, by both commercial and non-commercial researchers.

- Youth informed democracy with download (Group C): Study participants vote as described in Option 1 of Question 2 above. Under this model, researchers are allowed to download a copy of the data.

- Youth informed democracy without download (Group D): Study participants vote as above. Under this data governance model, data may only be accessed via a restricted server.

In order to mirror the experience in a typical study, participants were exposed to an informed consent specific to their data governance model and could choose to either join or not. They had no exposure to other potential governance models.

Participants randomized to Groups C and D, as well as those in Group A who selected the Democracy option for Choice 2, were prompted to vote on four questions covering (1) whether data could be used for for-profit endeavors, (2) whether researchers have to pay to use the data, (3) the types of research the data can be used for, and (4) how results should be shared (see Supporting Methods in S1 File for full list of questions and options).

**Analysis.** Participant preferences were quantified among consenting participants who were randomized to Group A. Significance was assessed via a Chi-square test for equal proportions for both Choice 1 and Choice 2 independently. A multinomial regression model was used to assess the significance of age, gender, country, and reported lived experience of mental health challenges using the *nnet* package [14] in R [15].

Participant acceptability was assessed by quantifying enrollment rates among participants randomized to Groups B, C or D. Logistic regression was used to assess significance of the governance model using the binomial family *glm* in R. In order to assess sensitivity to covariates, logistic regression models were fit with and without age, gender, country, and history of lived experience. Type II ANOVA was used to assess significance of categorical variables (gender, country, and history of lived experience) using the *car* package in R [16, 17]. The *emmeans* package [18] was used to contrast the effects of democracy versus researcher defined terms groups (Group C and D vs Group B) and server data access versus data download (Group D vs Groups B and C). We also performed analyses within each country. All analyses and visualizations were performed in R version 4.1.3 (2022-03-10).

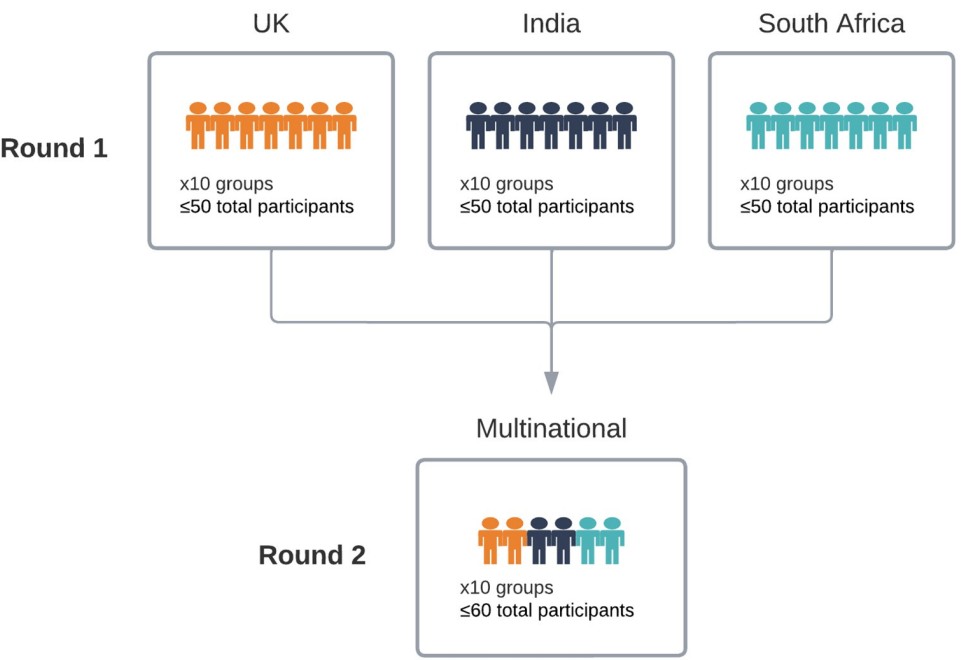

**Fig 3. Two-round, multi-country deliberation design.**

## Qualitative study arm

**Aims.** The aims of the qualitative study arm were (1) to identify the consensus data governance model(s) for an open—yet privacy preserving—global mental health databank from the perspective of pan-national youth (India, South Africa, UK) and (2) to understand the concerns, hopes, and expectations of pan-national youth for such a databank with regards to (a) return of value to youth participants and (b) youth participation in databank governance.

**Procedure.** We employed a public deliberation (also called deliberative democracy) design, which is a methodology for community engagement around complex ethical issues, often related to technology [19]. We convened young people ages 16–24 (UK) and 18–24 (India and South Africa) for two rounds of deliberative sessions (Fig 3), exploring in depth young people's feelings and experiences with sharing personal data, with a focus on mental health data, and their data governance preferences.

The first round of deliberation was led by in-country facilitators and enabled participants to become comfortable with communicating about data governance with individuals of a shared nationality. In this round, we aimed to engage 50 participants per site in up to 10 site-specific sessions, for a total of 150 participants. The second round invited youth participants who had attended the first round to participate in multinational sessions. We aimed to host approximately 10 sessions totaling 60 participants. Across all sessions, we sought stratification of younger (16–20) and older (21–24) participants as well as separation between participants co-enrolled in the quantitative study arm and those not co-enrolled ("naïve participants") in the quantitative study arm. Written consent was obtained from all participants prior to their first session. Qualitative sessions were conducted in English at all three sites and additionally conducted in two regional languages in India. At the South African site, participants were also encouraged to mix Indigenous South African languages with English to enable ease of communication, and translation was provided by facilitators as necessary. Session recordings were

transcribed verbatim and translated as needed. Consistent with Thorne, 2020 [20], quotes presented in this paper have been cleaned of *um*, *uh*, and *like* for clarity.

As is characteristic of public deliberation studies [19], participants were provided with detailed informational materials to inform their discussion. In addition to the data governance typology (Fig 1), we offered participants a set of four data governance models to provide use cases of the typology. The typology, four governance models, and their constituent components were produced through extensive iteration with our DUAG and YPAGs. We scripted and recorded two video modules and asked participants to watch the modules prior to their deliberative session. We also developed an interactive concept map (https://stroly.com/viewer/1620332775/) as an additional means of consuming the information [21]. All the informational materials used in this study are publicly available: https://www.synapse.org/#!Synapse:syn35371551.

Deliberative sessions were held remotely over Zoom due to the COVID-19 pandemic. Study team members at each site facilitated in-country sessions, and facilitators from each site were present at multinational sessions. Facilitators were largely PYAs and early career researchers. Facilitators utilized a facilitation guide (Supporting Methods in S1 File) and conducted "mock sessions" with YPAGs to prepare for data collection. Facilitators were directed to encourage contributions from all participants and to seek consensus for each of the seven data governance questions (Fig 1), asking participants to group data governance choices into *acceptable*, *unacceptable*, and *maybe* as a means of promoting discussion. Using exit survey and session data, we assessed the deliberative data to be of high quality [22] despite challenges posed by unstable internet access, limited bandwidth, and related technical infrastructure hurdles faced by some participants.

**Analysis.** Our outputs can be organized into deliberative outputs, which are the explicitly stated results of consensus-based reasoning during deliberative sessions, and analytical outputs, which are the deeper thematic findings derived from researcher analysis [23]. We organized consensus-based findings into graphical displays of acceptability and unacceptability. However, deliberative outputs included a great deal of variation. For this reason, we supplemented the explicit, consensus-based findings with common arguments made by participants regarding the data governance typology, which we defined as lines of argumentation made in very similar language by several different participants across multiple sites.

We utilized the framework method [24] for qualitative analysis, which we supplemented with a brief thematic analysis [25] procedure during the formation of original themes. Researchers at all three sites, including PYAs, were involved in the analysis, for which we used Miro [26] and Nvivo [27] software.

## Results

### Quantitative study arm

In total, 3575 young people consented to participation in the quantitative study arm (1034 from India, 932 from South Africa, 1609 from the UK) (S1 Table in S1 File). The participant pool were mostly young people with lived experience of mental health challenges (88%, 67%, 91% in India, South Africa, and the UK, respectively) and identified as women (87%, 79%, 64% in India, South Africa, and the UK, respectively). Nearly half of participants in the UK were aged 16 and 17 (48%); however, in India and South Africa the minimum age of participation was 18 (the local age of majority). Enrollment was well distributed across the 18–24 year old age groups. In order to assess enrollment rate, we also collected minimal demographic data from eligible potential participants who registered an account but did not complete the informed consent process or did not provide consent to join the app-based study. This

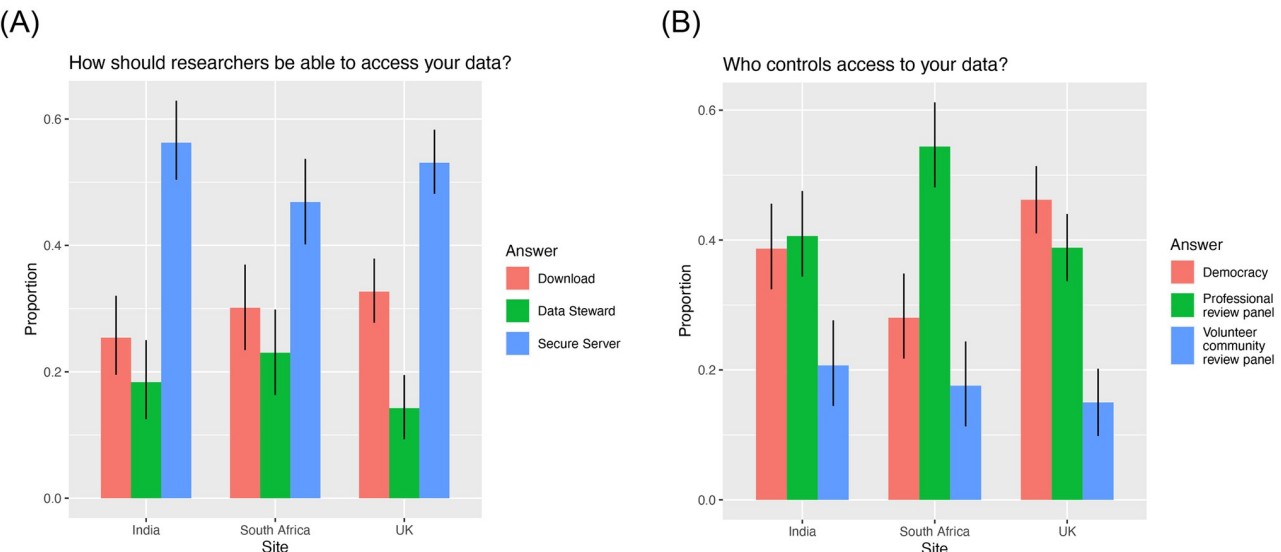

**Fig 4. Participant choice for (A) "*Who controls access to your data?*" and (B) "*How should researchers access your data?*" grouped by country for participants randomized to Group A.**

constituted an additional 1,409, 710 and 1,045 individuals from India, South Africa, and the UK, respectively.

**Participant preference.** Participants in *Group A*: *Participant Choice* strongly preferred Secure Server when given a choice about how researchers access their data (Chi-squared *p*-val < 2.2e-16) (Fig 4(A)). This was true across sites ($\hat{p}$(Secure Server) = 0.56, 0.47, 0.53, for India, South Africa, and UK, respectively). There was also no statistically significant difference by age, gender or lived experience of a mental health challenge (S2 Table in S1 File).

When given a choice about who controls access to the data, participants in India and UK showed a preference for Democracy or Professional Review Panel over Volunteer Review Panel (Chi-squared *p*-value = 9.499e-05 and 7.843e-15 for India and UK, respectively) (Fig 4 (B)). However, there was no statistically significant difference between Democracy and Professional Review Panel in either country (India 95% CI for Democracy = (0.32, 0.46) and for Prof. Review = (0.34, 0.48); UK 95% CI for Democracy = (0.41, 0.51) and for Prof. Review = (0.34, 0.44)). In contrast, South Africa showed a strong preference for Professional Review Panel ($\hat{p}$(Prof. Review) = 0.54, Chi-squared *p-value* = 6.168e-12) (Fig 4(B)). There was a modest effect of age (*p-value* = 0.05 and 0.05 for Democracy and Volunteer Review Panel, respectively, relative to Professional Review Panel) (S3 Table, S1 Fig in S1 File). For all three countries, older participants were less likely to choose Professional Review than younger participants.

**Participant acceptability.** In quantifying the difference in enrollment rates between participants in each group, we assessed whether democratic determination of access terms improves enrollment (comparison of enrollment rates between Group C vs Group B), and whether restricting data download additionally improves enrollment (Group D vs Group C). Enrollment between the three "acceptability" arms was not statistically significantly different (*p*-value = 0.218) (Fig 5, S4 Table in S1 File). This did not change by adding country, history of lived experience, age or gender to the model (*p*-value = 0.185), or by analyzing each country separately (*p*-value = 0.465, 0.627, and 0.056, for India, South Africa, and UK, respectively) (S5 Table in S1 File). However, in the UK there was a modest increase in enrollment in *Group D*: *Democracy + Server* over *Group B*: *Researcher Norms* (unadjusted p-value = 0.023, OR = 1.29,

## Consent Rate by Site and Consent Model

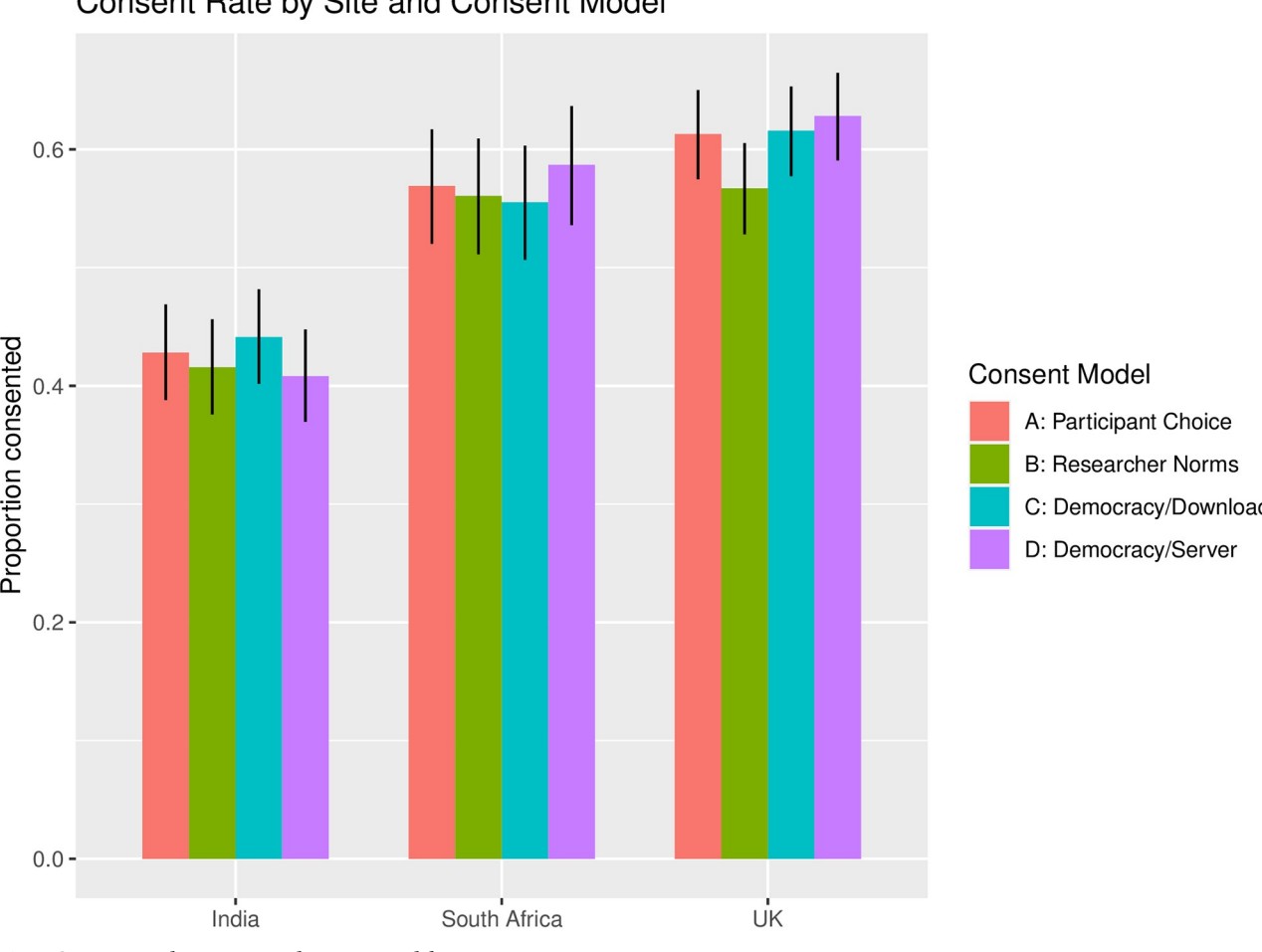

**Fig 5. Consent rate by country and consent model.**

95% CI = (1.04, 1.61)), which was not significant after Bonferroni correction for multiple testing. We saw similar patterns when contrasting the democracy versus researcher defined terms groups (Group C and D vs Group B) and server data access versus data download (Group D vs Groups B and C)(S6 Table in S1 File). On the whole, these results imply that while participants showed a preference for specific governance models, these models showed no substantial improvement in study enrollment.

Enrollment rate also did not significantly change by age (*p*-value = 0.884) or gender (p-value = 0.095), though it did show statistically significant differences by country (p-value < 2.2e-16), and by history of lived experience (p-value = 0.003) (S4 Table in S1 File). Notably, participants with history of lived experience were significantly more likely to enroll than participants who report no history (odds ratio (OR) = 1.26, 95% CI = (1.08, 1.47)).

**Democracy votes.** For those participants who chose or were randomized to democratic determination of criteria for accessing data, we assessed the votes on the four data use and access choices and, in most cases, found concordance across countries (S7 Table in S1 File). Participants from all countries preferred that their data not be used to make a profit and that commercial companies should have to pay to use the data. They also agreed that results should be shared for free with the world and in an easy-to-understand format for study participants. In contrast, there was disagreement about how the data could be used: while participants from

India and South Africa preferred that their data only be used for mental health research, participants from the UK showed a slight preference that their data be used for all types of health research.

## Qualitative study arm

In total, 143 young people participated in the qualitative study arm (46 from India, 52 from South Africa, 45 from the UK), with 61 of these individuals participating in the second round multinational sessions. Approximately half were co-enrolled in the quantitative study arm. Approximately half of participants were aged 16- (UK) or 18- (India and South Africa) to 20, and half were aged 21–24. Other demographic characteristics were not collected, but study teams were encouraged to recruit via demographically diverse networks. Recruitment of naïve participants was swift for all sites, with sites reporting the effectiveness of emails to personal networks and partner organizations. Naïve participants were invited to share the opportunity to participate in the qualitative study arm with their peers and in their social networks; this snowball strategy was also reported as effective. In contrast, the co-enrolled participants received an in-app pop-up notification with site specific links to join the study. Due to technical issues, some South African participants failed to receive this notification and alternative outreach methods (e.g., email) were employed.

**Deliberative findings.**   We identified a set of more and less acceptable data governance options (i.e., deliberative outputs [23]) and describe them across a gradient, with the Y-axis capturing both level of (un)acceptability and level of consensus (Fig 6). Accordingly, options that were widely agreed to be acceptable or widely agreed to be unacceptable to participants are clustered near the top and bottom of the figure, respectively. Options for which there was disagreement, often leading to a lack of consensus, are clustered near the middle of the figure; these options may have achieved consensus (as acceptable or unacceptable) in some sessions but not many. Graphical displays of preferences among in-country participants in India, South Africa, and the UK only are found in S2-S4 Figs in S1 File.

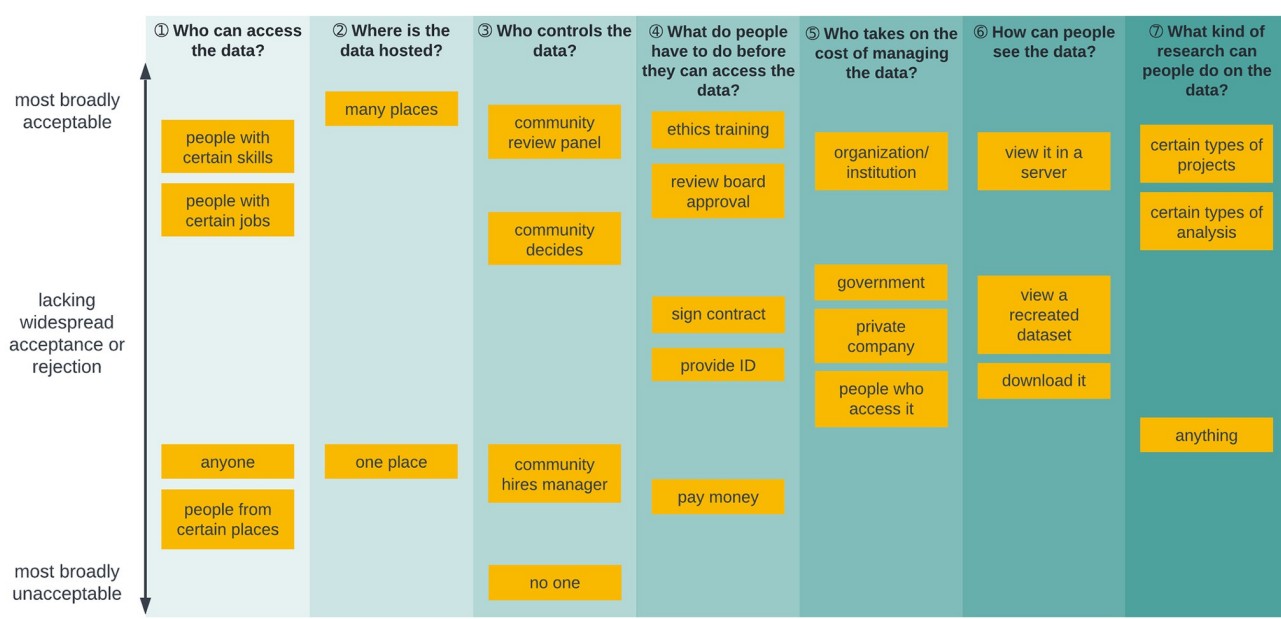

**Fig 6.  Aggregate deliberative outputs regarding acceptability and level of consensus.**

Common lines of argumentation made by participants in relation to these data governance options are presented in the Supporting Results in S1 File. These common lines of argumentation (below in **bold**) were at times contradictory. For example, when participants were asked to respond to Question 4: *What do people have to do before they can access the data*? we heard contradictory statements about whether having researchers sign contracts provided accountability, as illustrated below:

**A contract can be forged/one could deny that one signed it**.

*I'll go with signing the contract, because lately people be denying everything they've signed on* [. . .] *and mostly you can even win the case, because there's no proof that you have signed that thing; and it was you at that time with the document or contract.*

(South Africa Session 3)

**A contract offers accountability for misuse**.

*In case there is misuse of data*, [. . .] *you can go back to saying that, okay, this was the person, it's not any fake person*, [. . .] *if there are any legal repercussions or anything that would go on, this is how you conduct that.*

(India Session 4)

Far from a split between reasonable and unreasonable argumentation, the points at which common lines of argumentation contradict each other reflect the complexity of the choices—and their implications—with which participants were asked to grapple. It is perhaps unsurprising, then, that governance questions that asked participants to consider the role of money in research generated considerable debate and clear arguments for and against. For example, two contrasting lines of argumentation (below in **bold**) we heard in response to the question of researchers having to pay a fee to access the databank:

**Requiring researchers to pay money could be discriminatory**.

*So I think paying money is again, like completely unacceptable for me too, because, again,* [it would] *limit the data to only certain people* [who] *are probably some big giant people.*

(India Session 4)

**Requiring researchers to pay money demonstrates buy-in that protects against misuse**.

*It could just facilitate a better quality of research if universities or companies are having to pay for it.* [. . .] *It's more of an incentive to do better research.*

(Multinational Session 9)

In response to the question *Who takes on the cost of managing the data*?, we heard conflicting arguments (below in **bold**) about the effect of private funding.

**Having a private company fund the databank may benefit the company, but it benefits us in that it helps sustain the databank**.

*What I can imagine right now is pharma companies* [. . .], *which are more health centered you know, so, it would be great for them as well because they are investing in something which*

*is health related [. . .]. And it could also be great for the person you know, for all the researchers who are conducting research.*

(India Session 4)

**If a private company funds the databank, they may use it to make targeted ads**.

*I was also going to say, yeah, getting targeted ads based on the stuff that you look at, the last thing you really want is them being able to have access to private data about things like, for example, your mental health, because the last thing you want is something saying, "Oh, are you struggling from depression? Try Jack Daniel's whiskey."*

(UK Session 11)

Indeed, the divergence in argumentation regarding participants' preferences necessitated a deeper thematic analysis.

**Analytical findings.** The analytical framework used to represent the major themes that emerged in the study is displayed in Fig 7. Two overarching tensions were articulated by participants: a tension between controlling the data and access to it versus being unable to control the data or access to it, leading to a sense of what we called *resignation* among participants, or the feeling that no intervention could prevent nefarious actors doing what they wish with the data. A second tension arose between the benefits of data sharing versus the risks of data sharing. Whereas the latter tension is primarily concerned with the nature of *data*—its (in)security and its utility, the former tension is primarily concerned with how *people* in the databank ecosystem interact with it.

Beginning with the tension between the benefits of data sharing versus the risks of data sharing, participants envisioned that the findings derived from a mental health databank

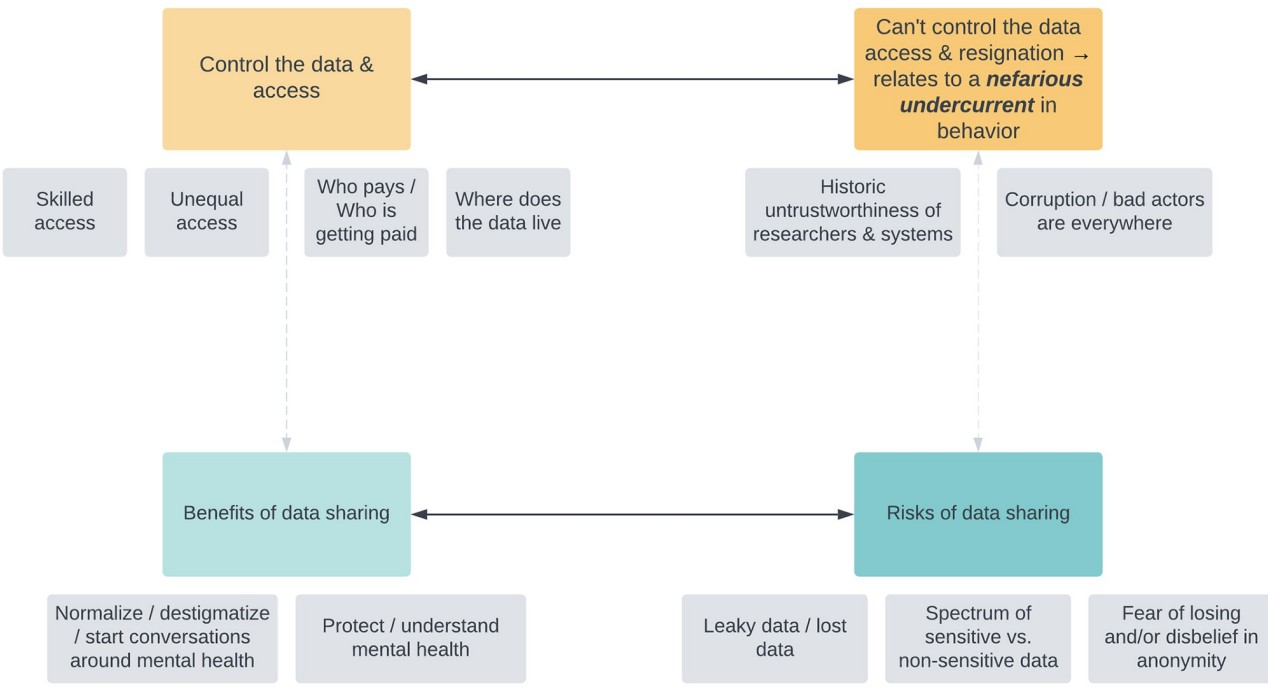

**Fig 7. Analytical framework of emergent themes and tensions derived from public deliberation data.**

would create community benefits such as normalizing, destigmatizing, or starting conversations around mental health:

> *There's still a lot of stigma around mental illness.* [. . .] *I will speak for where I am. Most people think it's witchcraft or it's just something that is weird. They don't have that mentality that it could be mental illness or something serious that can be mental–that needs professional help.* [. . .] *It would be safer to–for like, professionals in specific to have such information in order–in order for them to modify, change things. And for the public as well, for their personal growth and to know how to seek help when they–they actually need help.*

(Multinational Session 3)

Given the stakes of mental health stigma in their communities, the prospective benefit of more research in this area was recurrently expressed. However, participants weighed this benefit against the risks of data sharing, such as the concern that even anonymized data would carry a risk of re-identification in a sufficiently small sample:

> *I guess I wouldn't necessarily want to restrict it, but I think depending on the area you'd want to take into account how identifiable that data could become to someone in a certain region. Like if you were specifying someone was from a small Welsh town, and the people from the small Welsh town can access it.*

(UK Session 4)

Participants connected this concern of re-identification with a ubiquitous experience of user tracking in the era of big data: targeted advertising.

> *One misuse I can think of is that, if the private company gets access to people's IDs, etc.,* [. . .] *then using internet or social media they can reach people through advertising. For example, when we login to any app, then you get more advertisements of that app.*

(India Session 2, translated to English)

When deliberating what actors could do with re-identifiable data, participants grappled with the tension between controlling the data and access to it versus being unable to control the data or access to it. Consider this participant's response to a facilitator's question of what the least important data governance criteria was to them:

> *The least important, huh, okay, where the data is hosted. So whether it's hosted in one place or many places, as long as the right people have access to the data and can make a difference, it doesn't really matter.*

(South Africa Session 3)

This quote is a succinct articulation of why controlling access is ultimately a means to identify the "right people"; this participant indicates some ambivalence regarding the particular access conditions, so long as the parameters in place enable the "right people" to access the databank and "make a difference." Finding the "right people," however, is complicated by the presence of corruption and bad actors in everyday encounters with research systems. Even the gatekeepers of research ethics are potentially corruptible; consider this participant's response to the idea of ethics boards (such as institutional review boards) overseeing data use requests:

Participant: *Because with board approval, we might find out that the person that's trying to access the information has connection with some of the board members.*

Facilitator: *So you think maybe it might [. . .] be an avenue for people to kind of get in because of their connections rather than because of their merits?*

Participant: *Yeah, I feel like they wouldn't get the information because they qualified to but because they know someone.*

(South Africa Session 8)

When social environments are punctuated by bribery and nepotism, the "right people" are harder to find. Moreover, this dilemma is deeply rooted in the historic untrustworthiness of researchers and the research ecosystem as a whole:

*I generally think it's unacceptable for anyone to [. . .] access the data. Due to how we've seen that people sometimes manipulate data to use it for their own unsolicited or unscrupulous, you know, research. [. . .] So for example, [. . .] I think it was in the '70s that it was deemed that black people [. . .] were seen as people who couldn't get depression. And also, I think about 30 years ago it was seen that being gay or being homosexual is a mental illness. So, I think that certain things are due to all, how we've seen how history has played out.*

(Multinational Session 5)

Herein is the essential challenge of identifying the "right people": sometimes the wrong people—those researchers who have used their work to further racism or homophobia—have worn lab coats or have had the title "doctor." Indeed, while we saw participants share concerns about hackers or other face-evident bad actors, we also saw a thicker concern about historically harmful actors within respected research institutions:

*In theory, medical practitioners should inherently have had that ethics training built in, but I don't think that always happens in practice. There has been, you know, what is just blatantly unethical research done by full professionals with PhD. Andrew Wakefield is an obvious example. [. . .] Maybe this is just my outlook as a disabled person, but I don't feel like medical professionals can be inherently trusted just because that's their field, basically.*

(UK Session 1)

In calling attention to Wakefield, a disgraced former physician known for spuriously linking the MMR vaccine to autism [28], this participant articulates how research misconduct permeates out into the community and generates (understandable) mistrust.

These two tensions of the benefits of data sharing versus the risks of data sharing as well as the desire to control the data and access versus the feeling that one cannot control the data and access display the thickness of participants' perspectives, intersecting with history, culture, and technological capacity.

**Misconceptions.** We observed two common misconceptions from participants. Although we had specified in our informational materials that data contained in the databank would be anonymized, some participants' statements spotlight the shortcomings of our efforts to adequately explain this concept. Some participants' statements indicated that they believed a future databank would still contain identifiable information such as names, addresses, phone numbers, email addresses, date of birth, or national ID number. Distinct from the theme we called *fear of losing and/or disbelief of anonymity* (Fig 7), which reflects a sophisticated

understanding of how, in the era of big data, datasets can be combined to become re-identifiable, this misconception pertains to anonymization processes at a base level. This phenomenon is of note because if participants do believe that identifiable information is part of a databank, it may meaningfully influence their decisions for governing such a databank.

We saw this misconception in value statements that participants put forward:

*I think it depends whether our names are included or not. If our names are. . .included, I think that would obviously make it more uncomfortable.*

(Multinational Session 8)

*I think that people's identities shouldn't be revealed because it takes away the privacy.*

(South Africa Session 6)

And the level of researcher access that they envisioned:

*You could just search someone's name and date of birth and it would come up.*

(Multinational Session 9)

A second misconception present in the data was therapeutic misconception [29]. When participants described how they envisioned a global mental health databank, occasionally it appeared they were unaware that the primary intent of a research databank is to facilitate research (i.e., producing generalizable knowledge), not the provision of individual clinical care.

Participants at times envisioned a mental health app as a mobile psychiatric device rather than a data collection tool:

*What if sharing deep things helps me out like, getting out communicating my problems to the App helps me out. And those things are not known. Maybe I'm the only person who knows. And the App helps me to communicate with it. And I trusted it with my data.*

(South Africa Session 6)

*The App that is being created is a little like a kind of clinic, where people are able to search for answers to their questions or search for solutions. It feels like a kind of clinic. When we are sick, we go to a doctor and he charges us fees for the consult.*

(Multinational Session 2, translated to English)

The spread and persistence of these misconceptions highlight two key areas for further research and needed targeted intervention as we seek to engage participants in big data research.

## Discussion

The findings from the MindKind Study, consistent across both the quantitative and qualitative study arms, suggest that young people have strong preferences as to how their data should be governed and they are also keen to share their (anonymized) data for the benefit of empirical research. However, these preferences did not translate into any quantitative difference in the amount of data that was subsequently shared by participants with researchers in our app-mediated quantitative study arm.

Youth co-researchers and our professional research advisors were instrumental in helping us hone our research design to target areas of divergence regarding data governance preferences of youth and the researcher-driven "status quo." Youth participants preferred for the collected data to be primarily used to better understand and examine mental health problems or, at the most, health related topics. Concerns about monetization within the future global mental health databank were evidenced in both study arms. Most quantitative study arm participants who voted did not support their data being used to generate a profit and felt that there should be no access fees or fees only for commercial users (S7 Table in S1 File). Qualitative data corroborates these findings: participants articulated concern about fees for data use being a barrier to equitable access, highlighted inequalities perpetuated by the influence of money in research, and expressed generalized mistrust of private companies use of their data.

One strong preference expressed by youth across both arms of the study, which starkly contrasted with the opinions expressed by our professional research advisors during the study design, was the preference for data to be hosted on a secure server (i.e., that the data be "sandboxed") over data being available for download. While sandboxed data enclaves have been employed in several large databanks [30, 31], these workspaces currently require researchers to be somewhat computationally savvy, accessing and analyzing the data in scripting languages like R, Python, and SQL. This may be exclusionary to researchers who are accustomed to using user-interface (UI)-centric software and specialized, standalone packages for analysis. They typically also require a stable internet connection, which may be prohibitory for researchers in infrastructure-poor areas. Developing more inclusive sandboxed workspaces will be necessary to best serve the global researcher community. Of note, participants in the qualitative study were offered the option of data storage via *recreated dataset*, a term we used synonymously with synthetic dataset. This concept was difficult for participants to understand across all sites, with most participants ultimately stating a preference for hosting via secure server. If future public deliberation research probes the possibility of synthetic dataset use, particular attention should be paid to how this concept is explained to participants.

There were a few notable surprises. In contrast to the initial opinions expressed by our youth advisors and youth advisory panels, we did not find a strong preference for democratic determination of data access conditions (quantitative study arm Group A, Choice 2). In India and the UK, this choice was statistically equivalent to the professional review panel. Whereas in South Africa the professional review panel was strongly preferred. Qualitative data do not directly correspond to this finding, as qualitative study participants were not offered the option of a professional review panel per se. Within the qualitative study arm, South African participants uniquely raised the possibility that "review board members"—in the language that we used—may be corruptible. As is evident in the complexity of the thematic results, identifying the "right people" is a thorny process, and participants differed in who had the potential to be a bad actor.

Despite the strong governance preferences expressed by participants across the study, we did not observe a difference in enrollment rates under various governance models (which ranged from a traditional, researcher-friendly model that gave researchers more flexibility to how they access and use the data, to models that gave youth more control over how their data were used and accessed) within the quantitative study arm. We expected that traditional models would be less acceptable to youth participants than those which provide them with more control over how their data used or accessed. However, we found no statistically significant difference in enrollment rates across the models. It is important to note that this effect was not driven by the mediocre popularity of democratic determination of access conditions, as mentioned above. Even contrasting server data access versus data download (Group D vs Groups B & C) showed no statistically significant difference in enrollment rate (S6 Table in S1 File),

despite the strong preference for server access in both studies. We also note that in Group B, the fixed terms presented, namely unrestricted and commercial use, were highly unpopular among youth in both study arms. Despite the preferences expressed in both studies, those recruited to the quantitative study arm were equally likely to enroll and engage in the study regardless of the governance model presented.

In the qualitative study arm, while there were few areas of complete consensus for the governance of a prospective mental health databank, there were trends in acceptability. Participants broadly exhibited a desire for equitable access for researchers and institutions in low resource settings as well as a concern about the potential perverse influence of money in research. However, especially in regard to the latter, participants proposed multiple different—and at times contrasting—remedies for these areas of shared values. For instance, while some participants sought to eliminate private funding in databank management for fear of targeted ads, others made pragmatic arguments that private funding is necessary for databank sustainability. Indeed, the questions in the typology dealing with funding (questions 4 and 5) consistently elicited the most impassioned, value-based argumentation about the effect of financial structures on global communities. We recognized both in facilitation and analysis that discussions about funding had the richest texture. It was never merely a pragmatic discussion about fees for service—these conversations opened a window into participants' perspectives on who wins and who loses in the global economy. Participants expressed hope that the products of the databank would reduce stigma and improve mental healthcare. However, the depth of concern about corruption, breach of privacy, ad targeting, or undue profit was vast, rooted in the histories of researcher misconduct and cultural milieu of bad data actors. This pervasive nefarious undercurrent challenged even the research team, leading some facilitators to note that participants tried to "break the databank" (e.g., imagine ways in which actors could violate the terms of good faith usage) at all costs. Despite this, participants were active in their attempts to resolve areas of concern and, in exit surveys, exhibited a broad willingness to contribute their own data to a thoughtfully produced databank. In response to the exit survey inquiry "If a global mental health databank was created according to the specifications your group chose today, would you contribute data about yourself?", the response was 91% affirmative.

This study benefited from a complex design involving mixed methods and was conducted across sites on three continents, however we acknowledge the limitations of our findings. While we generally observed strong understanding of the informational materials in the qualitative study based on the robustness of the facilitated discussions, there is some evidence of therapeutic misconception among participants. Lemke, Halverson, and Ross had similar findings following a deliberation on biobank participation [32]. Like Lemke et al., our informational materials were thoroughly tested with youth co-researchers for relevance and comprehensibility prior to their use within the study. Particularly in the field of mental health, where participants may be acquainted with commercial apps, many of which claim therapeutic effect [33], we may need to be intentional in distinguishing research apps and their particular focus.

Language was a limitation across both arms of the study. The quantitative study was only available in English, which may have affected the population reached, particularly in India and South Africa. While the qualitative data obtained from non-English-speaking Indian participants made exceptionally rich contributions to our thematic understanding of participants' perspectives, this group experienced particular challenges comprehending concepts related to research and databanking. Barriers to understanding in this participant group, which was recruited intentionally for their sociocultural identities, mirror those found by public deliberation researchers with other marginalized groups [32]. While our informational materials were translated into regional languages by native speakers, additional attention is needed to ensure that concepts are presented in a way relevant to and understood by participants.

We chose to include participants ranging from the age of consent (i.e. those legally allowed to make decisions about study participation without the need for a parent/guardian) to the age of 24, in order to understand the choices that young people would make for themselves. This resulted in differences, by country, in the ages included (18–24 for India and South Africa; 16–24 for UK). However, we largely found no effect of age in our analysis, except for a minor increase in those choosing a professional review panel to control access to their data. In the qualitative study, we stratified groups by age in order to minimize power dynamics between participants.

Perhaps the most pervasive limitation of this study related to technology itself. The Android device requirement for the quantitative arm was a limiting factor affecting each site differently. Recruitment for the quantitative arm was hampered in the UK as substantially fewer eligible youth reported to have Android phones when approached at in-person student events (such as university orientation fairs). While Android phones have good market share in the UK [34], they tend to be substantially less popular among younger users [35], so quantitative results may not be generalizable to the rest of the UK population. In contrast, Android phones are highly represented in India and South Africa [36, 37], though the Android requirement is a potential limitation here as well. The device requirement may have also excluded the poorest participants of the study. Due to the high cost of cellular data in South Africa, we subsidized data usage for both the quantitative and qualitative arms; however, we still noted decreased participation, data loss, and connectivity issues (e.g., due to load shedding on the electricity grid) hampering participation. We also noted that a larger portion of participants were recruited through offline strategies. While the qualitative arm was not affected by the Android-specific limitations, the online nature of the qualitative sessions still posed a barrier to recruiting and retaining participants with limited network access. Notably, any future mental health databank developed based on our findings will need to confront the technology, language, and access issues in order to be representative with respect to economic, sociocultural, and language background.

## Supporting information

**S1 File.**
(PDF)

## Acknowledgments

The MindKind Consortium includes Faith Oluwasemilore Adeyemi[4,16], Patricia A. Areán[9], Emily Bampton[2], Elin A. Björling[3], Ljubomir Bradic[1], Anne-Marie Burn[4], Emma Grace Carey[4], Sonia Carlson[1], Pamela Y. Collins[9, 12], Tessa Concepcion[12], Meera Damji[6], Megan Doerr[1], *, Julia C. Dunbar[1], Mina Fazel[2], Blossom Fernandes[2], Gillian Finchilescu[17], Tamsin Ford[4, 14], Melvyn Freeman[10, 13], Isabell R. Griffith Fillipo[9,18], Jay Hodgson[1], Jasmine Kalha[5], Minal Karani[5], Michael R. Kellen[1], Christopher G. Kemp[19], Simthembile Lindani[6], Lara M. Mangravite[1], Carly Marten[1], Hedwick Masomera[6, 11], Felicia Mata-Greve[9], Emily Moore[1], Erin Mounts[1], Lakshmi Neelakantan[2], Larsson Omberg[1], Lisa Pasquale[1], Soumitra Pathare[5], Swetha Ranganathan[5], Nichole Sams[9], Erin Joy Scanlan[1], Himani Shah[5], Sotirios Short[13], Refiloe Sibisi[7,15], Solveig K. Sieberts[1], Stockard Simon[1], Sushmita Sumant[5], Christine Suver[1], Yanga Thungana[6], Meghasyam Tummalacherla[1], Chandre Van Vught[13], Jennifer Velloza[8, 12], and Zukiswa Zingela[11].

We acknowledge and thank the members of our Youth Group Advisory Panels in India, South Africa, and the UK, the International Youth Panel, the Global Youth Panel, and the Data Use Advisory Group for their contributions to this project.

[1] Sage Bionetworks, Seattle, Washington, United States of America

[2] Department of Psychiatry, University of Oxford, Oxford, Oxfordshire, United Kingdom

[3] Human Centered Design and Engineering, University of Washington, Seattle, Washington, United States of America

[4] Department of Psychiatry, University of Cambridge, Cambridge, Cambridgeshire, United Kingdom

[5] Centre for Mental Health Law & Policy, Indian Law Society, Pune, Maharashtra, India

[6] Department of Psychiatry, Walter Sisulu University, Eastern Cape, South Africa

[7] Activate Change Drivers ZA, Johannesburg, Gauteng, South Africa

[8] Department of Epidemiology & Biostatistics, University of California San Francisco, San Francisco, California, United States of America

[9] Department of Psychiatry and Behavioral Sciences, University of Washington, Seattle, Washington, United States of America

[10] University of Stellenbosch, Stellenbosch, Western Cape, South Africa

[11] Nelson Mandela University, Gqeberha, Eastern Cape, South Africa

[12] Department of Global Health, University of Washington, Seattle, Washington, United States of America

[13] Higher Health, Centurion, Gauteng, South Africa

[14] Cambridgeshire and Peterborough Foundation NHS Trust, Fulbourn, Cambridgeshire, United Kingdom

[15] University of Johannesburg, Johannesburg, Gauteng, South Africa

[16] Department of Psychology, University of Bath, Bath, United Kingdom

[17] Department of Psychology, University of the Witwatersrand, Johannesburg, Gauteng, South Africa

[18] CREATIV Lab, University of Washington, Seattle, Washington, United States

[19] Department of International Health, Johns Hopkins University, Baltimore, Maryland, United States

*Correspondences to: megan.doerr@sagebionetworks.org (MD)

## Author Contributions

**Conceptualization:** Anne-Marie Burn, Christine Suver, Jennifer Velloza, Pamela Y. Collins, Mina Fazel, Tamsin Ford, Melvyn Freeman, Soumitra Pathare, Zukiswa Zingela, Megan Doerr.

**Data curation:** Solveig K. Sieberts, Carly Marten, Meghasyam Tummalacherla.

**Formal analysis:** Solveig K. Sieberts, Carly Marten, Emily Bampton, Elin A. Björling, Anne-Marie Burn, Emma Grace Carey, Simthembile Lindani, Hedwick Masomera, Lakshmi Neelakantan, Swetha Ranganathan, Himani Shah, Refiloe Sibisi, Sushmita Sumant, Yanga Thungana, Meghasyam Tummalacherla, Megan Doerr.

**Funding acquisition:** Pamela Y. Collins, Mina Fazel, Tamsin Ford, Soumitra Pathare, Megan Doerr.

**Investigation:** Carly Marten, Emily Bampton, Anne-Marie Burn, Emma Grace Carey, Blossom Fernandes, Jasmine Kalha, Simthembile Lindani, Hedwick Masomera, Lakshmi Neelakantan, Swetha Ranganathan, Himani Shah, Refiloe Sibisi, Sushmita Sumant, Yanga Thungana.

**Methodology:** Solveig K. Sieberts, Carly Marten, Anne-Marie Burn, Blossom Fernandes, Jasmine Kalha, Lakshmi Neelakantan, Christine Suver, Jennifer Velloza, Mina Fazel, Tamsin Ford, Megan Doerr.

**Project administration:** Carly Marten, Anne-Marie Burn, Sonia Carlson, Blossom Fernandes, Jasmine Kalha, Lakshmi Neelakantan, Lisa Pasquale, Erin Joy Scanlan, Mina Fazel, Tamsin Ford, Melvyn Freeman, Soumitra Pathare, Zukiswa Zingela, Megan Doerr.

**Resources:** Megan Doerr.

**Software:** Solveig K. Sieberts, Sonia Carlson, Meghasyam Tummalacherla.

**Supervision:** Solveig K. Sieberts, Carly Marten, Elin A. Björling, Anne-Marie Burn, Blossom Fernandes, Jasmine Kalha, Christine Suver, Yanga Thungana, Pamela Y. Collins, Mina Fazel, Tamsin Ford, Melvyn Freeman, Soumitra Pathare, Zukiswa Zingela, Megan Doerr.

**Validation:** Carly Marten.

**Visualization:** Solveig K. Sieberts, Carly Marten, Elin A. Björling, Meghasyam Tummalacherla, Megan Doerr.

**Writing – original draft:** Solveig K. Sieberts, Carly Marten, Elin A. Björling, Megan Doerr.

**Writing – review & editing:** Solveig K. Sieberts, Carly Marten, Emily Bampton, Elin A. Björling, Anne-Marie Burn, Emma Grace Carey, Blossom Fernandes, Jasmine Kalha, Simthembile Lindani, Lakshmi Neelakantan, Swetha Ranganathan, Refiloe Sibisi, Christine Suver, Jennifer Velloza, Pamela Y. Collins, Mina Fazel, Tamsin Ford, Melvyn Freeman, Soumitra Pathare, Zukiswa Zingela, Megan Doerr.

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
