## [Decision Letter · Decision Letter 0]

7 Feb 2023

PONE-D-22-34513Young people’s data governance preferences for their mental health data: MindKind Study findings from India, South Africa, and the United KingdomPLOS ONE

Dear Dr. Doerr,

Thank you for submitting your manuscript to PLOS ONE. After careful consideration, we feel that it has merit but does not fully meet PLOS ONE’s publication criteria as it currently stands. Therefore, we invite you to submit a revised version of the manuscript that addresses the points raised during the review process.

Dear authors,

Thank you for submitting your work. Experience is highly relevant, novelty and important, but there are difficulties in finding several reviewers with the expertise to evaluate it.

In order not to delay the editorial process, and since we have an external review, I will take the figure of the second reviewer.

I ask you to clarify in the article the fact of selecting participants with ages in different ranges. Would that imply a bias? What did you do to mitigate this limitation?

Best regards,

We look forward to receiving your revised manuscript.

Kind regards,

André Ramalho, PhD

Academic Editor

PLOS ONE

Journal Requirements:

"In addition to the authors named above, the MindKind Consortium includes Faith Oluwasemilore Adeyemi, Patricia A. Areán, Ljubomir Bradic, Tessa Concepcion, Meera Damji, Julia C. Dunbar, Gillian Finchilescu, Isabell R. Griffith Fillipo, Jay Hodgson, Minal Karani, Michael R. Kellen, Christopher G. Kemp, Lara M. Mangravite, Felicia Mata-Greve, Emily Moore, Erin Mounts, Larsson Omberg, Nichole Sams, Sotirios Short, Stockard Simon, and Chandre Van Vught.

We acknowledge and thank the members of our Youth Group Advisory Panels in India, South Africa, and the UK, the International Youth Panel, the Global Youth Panel, and the Data Use Advisory Group for their contributions to this project. 

The MindKind Study was commissioned by the Mental Health Priority Area at Wellcome Trust. The funders had no role in study design, data collection and analysis, decision to publish, or preparation of the manuscript."

"The MindKind Study was commissioned by the Mental Health Priority Area at Wellcome Trust (https://wellcome.org/) from Sage Bionetworks (LMM and MD). The funders had no role in study design, data collection and analysis, decision to publish, or preparation of the manuscript."

"The authors have declared that no competing interests exist except for author Tamsin Ford.

Tamsin Ford declares: I have read the journal's policy and have the following competing interests: I consult to Place2Be, a third sector organization providing mental health support to children, parents and staff in Schools, and am the Vice Chair of the Association of Child and Adolescent Mental Health."

5. One of the noted authors is a group or consortium The MindKind Consortium. In addition to naming the author group, please list the individual authors and affiliations within this group in the acknowledgments section of your manuscript. Please also indicate clearly a lead author for this group along with a contact email address.

Additional Editor Comments:

Dear authors,

Thank you for submitting your work. Experience is highly relevant, novelty and important, but there are difficulties in finding several reviewers with the expertise to evaluate it.

In order not to delay the editorial process, and since we have an external review, I will take the figure of the second reviewer.

I ask you to clarify in the article the fact of selecting participants with ages in different ranges. Would that imply a bias? What did you do to mitigate this limitation?

Best regards,

Reviewers' comments:

Reviewer's Responses to Questions

**Comments to the Author**

1. Is the manuscript technically sound, and do the data support the conclusions?

Reviewer #1: Yes

2. Has the statistical analysis been performed appropriately and rigorously? 

Reviewer #1: Yes

3. Have the authors made all data underlying the findings in their manuscript fully available?

Reviewer #1: No

4. Is the manuscript presented in an intelligible fashion and written in standard English?

Reviewer #1: Yes

5. Review Comments to the Author

Reviewer #1: The authors present the results of an important project that describes the perspective of young individuals regarding their mental health data governance.

When the authors state "young participants", they include participants aged 16-24 in the UK and 18-24 in the other regions. Why didn't you selected the same age interval between countries? I do not see the benefits of including differential age groups between sites, please explain.

Please explain what is a "diverse youth" based on some variables.

Why the discrepancy on the % of participants allocated by social media advertisements between sites? Ellaborate the possible impacts on the results.

As you state in the discussion section, Android users represent only part of the population, possible selection bias?

6. PLOS authors have the option to publish the peer review history of their article (what does this mean?). If published, this will include your full peer review and any attached files.

Reviewer #1: No

---

## [Author Response · Author response to Decision Letter 0]

31 Mar 2023

We appreciate the reviewers’ supportive comments as well as the thoughtful concerns raised. We have addressed each one below.

Reviewer 1

When the authors state "young participants", they include participants aged 16-24 in the UK and 18-24 in the other regions. Why didn't you select the same age interval between countries? I do not see the benefits of including differential age groups between sites, please explain.

This study included participants from the age of consent (18 in India and South Africa, and 16 in the UK) to 24 years of age. We recognize that this introduced some differences across the countries examined, however we thought it important to include the youngest participants who were able to provide their own consent without the input of a parent/guardian. For the quantitative study, we did examine the effect of age as noted in the text. Age was not significant in the preferences related to ‘how researchers access the data’ nor in the analysis of participant acceptability. We did notice a modest effect (p-val =0.05) for the choice of ‘who controls access to the data’ with younger participants being more likely to choose professional review panel.

For the qualitative study, we included participants of the same age ranges to best produce results that corresponded to the quantitative study. We separated groups by younger (16-20) and older (21-24) participants to minimize power dynamics between participants, which we detailed in the Materials and methods as follows: “Across all sessions, we sought stratification of younger (16-20) and older (21-24) participants…”

We have added the following text to the discussion:

“We chose to include participants ranging from the age of consent (i.e. those legally allowed to make decisions about study participation without the need for a parent/guardian) to the age of 24, in order to understand the choices that young people would make for themselves. This resulted in differences, by country, in the ages included (18-24 for India and South Africa; 16-24 for UK). However, we largely found no effect of age in our analysis, except for a minor increase in those choosing a professional review panel to control access to their data. In the qualitative study, we stratified groups by age in order to minimize power dynamics between participants.”

Please explain what is a "diverse youth" based on some variables.

Thank you for bringing this to our attention. We wrote “PYAs sought to ensure members represented diverse youth from their countries based on geographic region, race/ethnicity, gender, language, and lived experience of mental health concerns.” We recognize that this language was unclear as to how the seeking of diversity was operationalized in our Young People's Advisory Groups (YPAGs). We have modified this language as follows: “Each YPAG was composed of 12-16 young people, and PYAs sought to ensure maximal heterogeneity within the groups with respect to geographic region, race/ethnicity, gender, languages spoken, and lived experience of mental health concerns. Study teams also worked with community partner organizations to ensure the representation of YPAG members from marginalized social groups.”

Why the discrepancy on the % of participants allocated by social media advertisements between sites? Elaborate the possible impacts on the results.

Each site employed a variety of different approaches to recruitment, however unique attributes of each country likely affected which approaches were successful. For example, the high cost of data, load shedding, etc in South Africa likely contributed to online recruitment being less successful here. We have a forthcoming publication delving into this topic so have not discussed it in-depth in this manuscript. We have added the following note in the limitations section:

“We also noted that a larger portion of participants were recruited through offline strategies.”

As you state in the discussion section, Android users represent only part of the population, possible selection bias?

Yes, we have noted this in the discussion “While Android phones have good market share in the UK [34], they tend to be substantially less popular among younger users [35], so quantitative results may not be generalizable to the rest of the UK population.” We have added “In contrast, Android phones are highly represented in India and South Africa [36, 37], though the Android requirement is a potential limitation here as well.” to make clear that the Android requirement affected representativeness in all three countries. 

Reviewer 2

I ask you to clarify in the article the fact of selecting participants with ages in different ranges. Would that imply a bias? What did you do to mitigate this limitation?

This study included participants from the age of consent (18 in India and South Africa, and 16 in the UK) to 24 years of age. We recognize that this introduced some differences across the countries examined, however we thought it important to include the youngest participants who were able to provide their own consent without the input of a parent/guardian. For the quantitative study, we did examine the effect of age as noted in the text. Age was not significant in the preferences related to ‘how researchers access the data’ nor in the analysis of participant acceptability. We did notice a modest effect (p-val =0.05) for the choice of ‘who controls access to the data’ with younger participants being more likely to choose professional review panel.

For the qualitative study, we included participants of the same age ranges to best produce results that corresponded to the quantitative study. We separated groups by younger (16-20) and older (21-24) participants to minimize power dynamics between participants, which we detailed in the Materials and methods as follows: “Across all sessions, we sought stratification of younger (16-20) and older (21-24) participants…”

We have added the following text to the discussion:

“We chose to include participants ranging from the age of consent (i.e. those legally allowed to make decisions about study participation without the need for a parent/guardian) to the age of 24, in order to understand the choices that young people would make for themselves. This resulted in differences, by country, in the ages included (18-24 for India and South Africa; 16-24 for UK). However, we largely found no effect of age in our analysis, except for a minor increase in those choosing a professional review panel to control access to their data. In the qualitative study, we stratified groups by age in order to minimize power dynamics between participants.”

---

## [Editor Report · Decision Letter 1]

4 Apr 2023

Young people’s data governance preferences for their mental health data: MindKind Study findings from India, South Africa, and the United Kingdom

PONE-D-22-34513R1

Dear Dr. Doerr,

We’re pleased to inform you that your manuscript has been judged scientifically suitable for publication and will be formally accepted for publication once it meets all outstanding technical requirements.

Kind regards,

André Ramalho, PhD

Academic Editor

PLOS ONE

---

## [Editor Report · Acceptance letter]

10 Apr 2023

PONE-D-22-34513R1 

Young people’s data governance preferences for their mental health data: MindKind Study findings from India, South Africa, and the United Kingdom 

Dear Dr. Doerr:

I'm pleased to inform you that your manuscript has been deemed suitable for publication in PLOS ONE. Congratulations! Your manuscript is now with our production department. 

Kind regards, 

on behalf of

Prof. Dr. André Ramalho 

Academic Editor

PLOS ONE